# Sex Differences in Motor and Non-Motor Symptoms among Spanish Patients with Parkinson’s Disease

**DOI:** 10.3390/jcm12041329

**Published:** 2023-02-07

**Authors:** Diego Santos-García, Ariadna Laguna, Jorge Hernández-Vara, Teresa de Deus Fonticoba, Carlos Cores Bartolomé, Maria J. Feal Painceiras, Maria Cristina Íñiguez-Alvarado, Iago García Díaz, Silvia Jesús, Maria Teresa Boungiorno, Lluís Planellas, Marina Cosgaya, Juan García Caldentey, Nuria Caballol, Ines Legarda, Iria Cabo, Lydia López Manzanares, Isabel González Aramburu, Maria A. Ávila Rivera, Víctor Gómez Mayordomo, Víctor Nogueira, Víctor Puente, Julio Dotor García-Soto, Carmen Borrué, Berta Solano Vila, María Álvarez Sauco, Lydia Vela, Sonia Escalante, Esther Cubo, Francisco Carrillo Padilla, Juan C. Martínez Castrillo, Pilar Sánchez Alonso, Maria G. Alonso Losada, Nuria López Ariztegui, Itziar Gastón, Jaime Kulisevsky, Manuel Menéndez González, Manuel Seijo, Javier Rúiz Martínez, Caridad Valero, Mónica Kurtis, Jessica González Ardura, Ruben Alonso Redondo, Carlos Ordás, Luis M. López Díaz, Darrian McAfee, Pablo Martinez-Martin, Pablo Mir

**Affiliations:** 1Department of Neurology, CHUAC, Complejo Hospitalario Universitario de A Coruña, c/ As Xubias 84, 15006 A Coruña, Spain; 2Neurodegenerative Diseases Research Group, Vall d’Hebron Research Institute (VHIR), 08035 Barcelona, Spain; 3CIBERNED (Centro de Investigación Biomédica en Red Enfermedades Neurodegenerativas), 19171 Madrid, Spain; 4CHUF, Complejo Hospitalario Universitario de Ferrol, 15405 Ferrol, Spain; 5Unidad de Trastornos del Movimiento, Servicio de Neurología y Neurofisiología Clínica, Instituto de Biomedicina de Sevilla, Hospital Universitario Virgen del Rocío/CSIC/Universidad de Sevilla, 41013 Seville, Spain; 6Hospital Universitari Mutua de Terrassa, 08221 Terrassa, Spain; 7Clínica del Pilar, 50006 Barcelona, Spain; 8Hospital Clínic de Barcelona, 08036 Barcelona, Spain; 9Centro Neurológico Oms 42, 07003 Palma de Mallorca, Spain; 10Consorci Sanitari Integral, Hospital Moisés Broggi, 08970 Sant Joan Despí, Spain; 11Hospital Universitario Son Espases, 07120 Palma de Mallorca, Spain; 12Complejo Hospitalario Universitario de Pontevedra (CHOP), 36001 Pontevedra, Spain; 13Hospital Universitario La Princesa, 28006 Madrid, Spain; 14Hospital Universitario Marqués de Valdecilla, 39008 Santander, Spain; 15Consorci Sanitari Integral, Hospital General de L’Hospitalet, L’Hospitalet de Llobregat, 08906 Barcelona, Spain; 16Hospital Universitario Clínico San Carlos, 28040 Madrid, Spain; 17Hospital Da Costa, 27880 Burela, Spain; 18Hospital del Mar, 08003 Barcelona, Spain; 19Hospital Universitario Virgen Macarena, 41009 Sevilla, Spain; 20Hospital Infanta Sofía, 28702 Madrid, Spain; 21Institut d’Assistència Sanitària (IAS)–Institut Català de la Salut, 17190 Girona, Spain; 22Hospital General Universitario de Elche, 03203 Elche, Spain; 23Fundación Hospital de Alcorcón, 28922 Madrid, Spain; 24Hospital de Tortosa Verge de la Cinta (HTVC), 43500 Tortosa, Spain; 25Complejo Asistencial Universitario de Burgos, 09006 Burgos, Spain; 26Hospital Universitario de Canarias, 38320 San Cristóbal de la Laguna, Spain; 27Hospital Universitario Ramón y Cajal, IRYCIS, 28034 Madrid, Spain; 28Hospital Universitario Puerta de Hierro, 28222 Madrid, Spain; 29Hospital Álvaro Cunqueiro, Complejo Hospitalario Universitario de Vigo (CHUVI), 36312 Vigo, Spain; 30Complejo Hospitalario de Toledo, 45007 Toledo, Spain; 31Complejo Hospitalario de Navarra, 31008 Pamplona, Spain; 32Hospital de Sant Pau, 08025 Barcelona, Spain; 33Hospital Universitario Central de Asturias, 33011 Oviedo, Spain; 34Hospital Universitario Donostia, 20014 Donostia, Spain; 35Hospital Arnau de Vilanova, 25198 Valencia, Spain; 36Hospital Ruber Internacional, 28034 Madrid, Spain; 37Hospital de Cabueñes, 33394 Gijón, Spain; 38Universitario Lucus Augusti (HULA), 27003 Lugo, Spain; 39Hospital Rey Juan Carlos, 28933 Madrid, Spain; 40Department of Neurosurgery, University of Maryland School of Medicine, Baltimore, MD 21201, USA

**Keywords:** motor, non-motor symptoms, gender, Parkinson’s disease, quality of life, sex

## Abstract

Background and objective: Sex plays a role in Parkinson’s disease (PD) mechanisms. We analyzed sex difference manifestations among Spanish patients with PD. Patients and Methods: PD patients who were recruited from the Spanish cohort COPPADIS from January 2016 to November 2017 were included. A cross-sectional and a two-year follow-up analysis were conducted. Univariate analyses and general linear model repeated measure were used. Results: At baseline, data from 681 PD patients (mean age 62.54 ± 8.93) fit the criteria for analysis. Of them, 410 (60.2%) were males and 271 (39.8%) females. There were no differences between the groups in mean age (62.36 ± 8.73 vs. 62.8 ± 9.24; *p* = 0.297) or in the time from symptoms onset (5.66 ± 4.65 vs. 5.21 ± 4.11; *p* = 0.259). Symptoms such as depression (*p* < 0.0001), fatigue (*p* < 0.0001), and pain (*p* < 0.00001) were more frequent and/or severe in females, whereas other symptoms such as hypomimia (*p* < 0.0001), speech problems (*p* < 0.0001), rigidity (*p* < 0.0001), and hypersexuality (*p* < 0.0001) were more noted in males. Women received a lower levodopa equivalent daily dose (*p* = 0.002). Perception of quality of life was generally worse in females (PDQ-39, *p* = 0.002; EUROHIS-QOL8, *p* = 0.009). After the two-year follow-up, the NMS burden (Non-Motor Symptoms Scale total score) increased more significantly in males (*p* = 0.012) but the functional capacity (Schwab and England Activities of Daily Living Scale) was more impaired in females (*p* = 0.001). Conclusion: The present study demonstrates that there are important sex differences in PD. Long-term prospective comparative studies are needed.

## 1. Introduction

Previously published data indicated that there are sex differences in many features of Parkinson’s disease (PD) [1,2]. Incidence and prevalence of PD are between 1.5 and 2 times higher in men than in woman and disease onset in women is slightly later than in men [3,4]. Clinically, males have more rigidity and speech problems, whereas females seem to have a greater prevalence of dyskinesia and motor and non-motor response fluctuations but more mild motor deterioration [5,6,7]. Regarding non-motor symptoms (NMSs), some studies have observed that symptoms such as fatigue, depression, anxiety, and sleep problems are more frequent in females, whereas other NMSs such as drooling, diurnal somnolence, urinary and sexual dysfunction, and cognitive problems are more prevalent in males [8,9,10]. However, the results are not consistent due to the great heterogeneity between the studies [2], including aspects such as the sample (e.g., differences in age, disease duration, or race), the type of evaluation conducted, or the follow-up period. Although several findings indicate that estrogens may play a role in PD and explain some sex differences, other mechanisms such as gene expression, neuroinflammation, oxidative stress, and lifestyle factors could be involved [11,12]. From a practical point of view, precise identification of the sex differences is important to tailor treatment, predict outcomes, and meet other individual and social needs in women and men with PD. In particular, it may be important to know sex differences in a specific population (i.e., in a province and/or a country) with the intention of developing global care strategies that can improve health care [13]. 

Although some epidemiological studies about PD have been conducted in Spain, the differences in motor and NMSs and disease progression between males and females in Spanish PD patients are unclear [14,15]. The aim of the present study was to analyze the main sex differences among Spanish patients with PD using data obtained from the COPPADIS cohort [16], the largest cohort of patients with PD studied in Spain to date. Specifically, we analyzed the differences between both sexes in terms of motor and NMSs, quality of life (QoL), autonomy, and disease progression.

## 2. Material and Methods

Patients with PD who were recruited from 35 centers of 14 Autonomous Communities of Spain from the COPPADIS cohort [16] from January 2016 to November 2017 were included in the present study. Methodology about the COPPADIS-2015 study can be consulted in https://bmcneurol.biomedcentral.com/articles/10.1186/s12883-016-0548-9 (accessed on 31 January 2023) [17]. This is a multi-center, observational, longitudinal-prospective, five-year follow-up study designed to analyze disease progression in a Spanish population of PD patients. All patients included were diagnosed according to UK PD Brain Bank criteria [18].

Specifically, the present study is a post hoc analysis with two parts. The first part included a cross-sectional analysis comparing data collected from the baseline visit (V0) regarding sex. The second was a longitudinal prospective analysis to compare the change between males and females in PD characteristics from the baseline visit (V0) to a follow-up visit at 24 months ± 1 month (V24M). For some variables, data about baseline visit, 12 months ± 1 month (V12M), and V24M were available. 

### PD Patient Assessment

In PD subjects, information on sociodemographic aspects, factors related to PD, comorbidity, and treatment was collected at baseline (visit V0) and at 24 months ± 1 month (visit V24M). V0 and V24M evaluations included (1) motor assessment (Hoenh and Yahr [HY], Unified Parkinson’s Disease Rating Scale [UPDRS] part III and part IV [19], Freezing of Gait Questionnaire [FOGQ]), (2) NMSs (Non-Motor Symptoms Scale [NMSS], Parkinson’s Disease Sleep Scale [PDSS], Visual Analog Scale-Pain [VAS-Pain], Visual Analog Fatigue Scale [VAFS]), (3) cognition (PD-CRS), (4) mood and neuropsychiatric symptoms (BDI-II, Neuropsychiatric Inventory [NPI], Questionnaire for Impulsive-Compulsive Disorders in Parkinson’s Disease-Rating Scale [QUIP-RS]), (5) disability (Schwab and England Activities of Daily Living Scale [ADLS]), and (6) health-related (the 39-item Parkinson’s disease Questionnaire [PDQ-39]) and global QoL (the EUROHIS-QOL 8-item index EUROHIS-QOL8]) [17]. At V12M, only the UPDRS, NMSS, and ADLS were assessed. In all the scales/questionnaires, a higher score indicates a more severe affectation except PD-CRS, PDSS, ADLS, and EUROHIS-QOL8, which is the opposite. Total score of the PDQ-39 (PDQ-39SI) and each domain of the NMSS and the PDQ-39 was expressed as a percentage: (score/total score) × 100 [20]. In patients with motor fluctuations, the motor assessment was made during the OFF state (without medication during the last 12 h) and during the ON state. The assessment was only performed without medication in patients without motor fluctuations. Non-motor assessment was conducted after taking medication. 

The presence of motor fluctuations and dyskinesia was assessed according to the UPDRS-IV (items 39 and 32, respectively) [19]. FOG was defined according to the FOGQ-item 3 [21,22]. Motor phenotype was calculated based on a previously published formula [23,24]. Very severe NMS burden was defined as a NMSS total score >70 [20]. Patient with a score <81 on the PD-CRS were considered as with cognitive impairment [25,26]. Patients were classified as with major depression according to the DSM-IV criteria [27,28]. Impulse control disorder (ICD) (pathological gambling, compulsive shopping, hypersexuality, and compulsive eating behavior) and compulsive behavior (CB) (punding, hobbyism, and dopamine dysregulation syndrome) were defined according to the previously published cutoff points of the QUIP-RS: gambling ≥6, buying ≥8, sex ≥8, eating ≥7, hobbyism–punding ≥7 [28,29]. For dopamine dysregulation syndrome, we accounted for the investigator’s criterion since an established cutoff does not exist [30]. 

## 3. Data Analysis

Data were processed using SPSS 20.0 for Windows. Only PD patients with data available in both visits at V0 and at V24M were considered valid for the longitudinal analysis. For comparisons between males and females, Student’s *t*-test, the Mann–Whitney U test, chi-square test, or Fisher’s test, were used as appropriate (distribution for variables was verified by one-sample Kolmogorov–Smirnov test).

General linear model (GLM) repeated measure were used to test for changes in various scores (motor; NMS; QoL; autonomy for ADL) over time (V24M vs. V0 and/or V24M vs. V12M and V12M vs. V0) separately in each group (men vs. women) and test for differences between groups over time. Age at baseline, and levodopa equivalent daily dose (LEDD [31]) at baseline and at V24M were included as covariates. In the latter models, an interaction for visit and group was tested before testing for a group difference over time. Cohen’s d formula was applied for measuring the effect size in both groups, males and females. It was considered: <0.2–Negligible; 0.2–0.49–Small; 0.50–0.79–Moderate; ≥0.80–Large. The Bonferroni method was used to *p* value correction for multiple comparisons and a *p* ≤ 0.002 was considered significant.

## 4. Results

At baseline, data from 681 PD patients (mean age 62.54 ± 8.93) were valid for the analysis. Of them, 410 (60.2%) were males and 271 (39.8%) females. There were no differences between both groups (males vs. females) in mean age (62.36 ± 8.73 vs. 62.8 ± 9.24; *p* = 0.297) or in the time from symptoms onset (5.66 ± 4.65 vs. 5.21 ± 4.11; *p* = 0.259). Compared to males, to be single (11.1% vs. 4.4%; *p* < 0.0001) and living alone (15.6% vs. 7.6%; *p* < 0.0001) was more frequent in females (Table 1). To be taking antidepressant agents (33.3% vs. 18.1%; *p* < 0.0001), benzodiazepines (23.2% vs. 11.2%; *p* < 0.0001), and analgesics (31% vs. 19.8%; *p* = 0.002) was more frequent in females as well. Up to 91.1% of females were non-drinkers compared to 71.1% of males (*p* < 0.0001). Regarding antiparkinsonian treatment, LEDD (607.79 ± 432.61 vs. 503.71 ± 381.49; *p* = 0.002) was higher in males than in females.

Although no differences in the score of the motor scales (HY, UPDRS-III, UPDRS-IV, FOGQ) were observed between sexes (Table 2), sub-scores of the UPDRS-III (OFF) detected a greater severity in hypomimia, speech, and rigidity (*p* < 0.0001 for all analysis) in males compared to females (Table 3 and Figure 1A). Regarding NMSs, females had a higher score on the BDI-II (9.95 ± 7.89 vs. 7.95 ± 6.84; *p* = 0.001) and major depression almost doubled (22.9% vs. 12%; *p* < 0.0001) in females when compared to males. The scores on the VAS about pain and physical fatigue were higher also in females (Table 2). Although the frequency of ICD (12.2%) and CB (9.3%) was similar in both sexes (Table 3), hypersexuality was clearly more frequent in males (7.6% vs. 0.4%; *p* < 0.0001). There were no differences between males and females in the NMSS total score (43.49 ± 36.89 vs. 47.94 ± 39.52; *p* = 0.208) but by domains, a trend in significance was detected for a greater burden in sleep/fatigue (18.03 ± 16.72 vs. 15.35 ± 15.63; *p* = 0.041), mood/apathy (13.78 ± 17.96 vs. 9.79 ± 15.19; *p* = 0.003), and miscellaneous (16.98 ± 16.71 vs. 13.56 ± 14.49; *p* = 0.010) in females whereas in urinary symptoms (22.48 ± 22.22 vs. 20.29 ± 22.84; *p* = 0.047) in males (Table 3 and Figure 1B). The results regarding global, fronto-subcortical, and posterior cortical cognitive function as a whole were similar in both groups (Table 3).

The perception of health-related QoL (PDQ-39SI, 18.88 ± 13.76 vs. 16.05 ± 13.33; *p* = 0.002) was worse in females. “Emotional well-being” (25.64 ± 20.97 vs. 18.6 ± 18.72; *p* < 0.0001) and “Pain and discomfort” (33.76 ± 24.31 vs. 21.46 ± 20.2; *p* < 0.0001) were the domains of the PDQ-39 with the clearest differences with a greater severity in females (Table 3 and Figure 2A), but “Communication” was worse in males (11.99 ± 16.24 vs. 7.74 ± 13.52; *p* < 0.0001). With regard to the EUROHIS-QOL8, a trend in significance was detected for a worse perception in females in the total score (3.71 ± 0.56 vs. 3.82 ± 0.53; *p* = 0.009) and in “Quality of life” (3.73 ± 0.72 vs. 3.86 ± 0.7; *p* = 0.014), “Health status” (3.07 ± 0.87 vs. 3.23 ± 0.87; *p* = 0.020), and “Energy” (3.65 ± 0.88 vs. 3.84 ± 0.77; *p* = 0.004) (Table 3 and Figure 2B). No differences were detected between males and females in terms of autonomy for activities of daily living (ADL) (Table 2).

Although the score on the UPDRS-III increased significantly from V0 to V24M in both groups, males and females, the difference between them was not significant (*p* = 0.554) (Table 4 and Figure 3). However, hypomimia and speech impairment from V0 to V24M in males but not in females was highly significant (*p* < 0.0001) (Table 4). Collectively, we observed an impairment of NMS burden (NMSS total score) after the two-year follow-up in both groups (Table 4 and Figure 3), with a trend in significance in males than in females (Cohen’s d, 0.43 vs. 0.25; difference between groups, *p* = 0.012). From V0 to V24M, functional capacity for ADL (ADLS) was impaired in males (from 89.12 ± 9.08 to 85.23 ± 12.58; *p* < 0.0001) and females (from 87.91 ± 11.42 to 82.76 ± 14.69; *p* < 0.0001), but the effect was greater in females than in males (*p* = 0.001; Table 4). Although LEDD tended to be higher in males than in females in all visits (*n* = 470; V0, 624.37 ± 434.36 vs. 514.88 ± 379.45 [*p* = 0.006]; V12M, 702.06 ± 401.64 vs. 621.96 ± 432.72 [*p* = 0.010]; V24M, 808.55 ± 460.97 vs. 726.31 ± 460.68 [*p* = 0.017]) (Figure 3), similar results were detected after dividing LEDD by weight (males vs. females, *n* = 450; V0, 7.81 ± 5.52 vs. 7.74 ± 6.07 [*p* = 0.550]; V12M, 8.71 ± 4.97 vs. 9.47 ± 6.96 [*p* = 0.782]; V24M, 10.94 ± 6.16 vs. 10.58 ± 7.44 [*p* = 0.166]). 

Finally, after comparison of all data of baseline visit (variables included in Table 1 and Table 2) between patients who were assessed at V24M (*n* = 496) and those patients who were lost to follow-up at V24M (*n* = 185), no significant differences were detected (only *p* < 0.005 in the PDSS total score, with a worse quality of sleep in patients who lost the follow-up; 108.87 ± 32.28 vs. 117.02 ± 24.47; *p* = 0.019).

## 5. Discussion

The present study analyzes the differences between Spanish males and females with PD in motor features, NMSs, QoL, and autonomy for ADL. This is the first study carried out in Spain that analyzes sex differences in a large cohort and one of the largest studies with longitudinal data and detailed evaluations carried out to date. Depression, fatigue, pain, and a worse QoL were more frequent in females whereas hypomimia, speech problems, and hypersexuality were associated with the male sex. In the short-term, males showed a tendency to develop a greater NMS burden impairment, but females deteriorated more in their functional capacity to perform their ADL. To know the differences in PD progression by sex is important because it allows for more personalized medicine, considering age, race, sex, and cultural context has become the vanguard of delivery of care [32].

Multiple studies analyzing sex differences in PD have been published to date. Many of them have been case-control, retrospective, or cross-sectional studies, and lack consistent findings [2]. In our cross-sectional analysis, we detected many differences between males and females but not in age and time from symptoms onset, so both groups were appropriate to compare. Aligning with our results, previous studies observed that women are more likely to live alone; however, while a spouse or partner is the most likely individual to serve as a caregiver, homebound women are more likely to be single or widowed compared to men [33]. Additionally, in our cohort, primary education, to receive more drugs for other diseases, and assistance to a PD association tended to be more frequent in women, as opposed to harmful habits (smoking and alcohol consumption), which were significantly more frequent in males. Although lifestyle factors are important in PD [34], reviews about the differences in PD by sex lack information about these factors and focus more other aspects [1,2,11,35,36,37,38,39]. 

Sex-related differences have been documented in several aspects of PD, including motor features, response to levodopa, NMSs, and QoL, among others [1,2,4,5,6,7,8,9,10,11,35,36,37,38,39]. Motor symptoms emerge later in women, including specific characteristics such as reduced rigidity [6,40,41] but also more risk of falling [11,41], as we found. On the contrary, speech problems, in line with our findings, freezing of gait, and drooling are more associated with male sex [1,2,11,42,43]. Although the frequency of hypomimia seems to be no different between sexes [44], we identified it as more severe in males. In fact, both axial signs, hypomimia, and speech problems were impaired after a two-year follow-up in males but not in females. Other axial symptoms such as camptocormia have been related to male sex as well [45]. Although the score on the UPDRS-III during the OFF state was higher in males from our cohort in all visits compared to females, the differences were not significant, nor was motor progression in both groups at the follow-up. Using data from the PPMI cohort, Picillo et al. [5] detected in a five-year longitudinal analysis that men had more longitudinal progression in clinician-assessed motor features in the ON medication state but a similar increase over time of MDS-UPDRS part III OFF scores in both sexes. Other studies suggests that women with PD have milder motor symptoms compared to men with PD [46]. However, Abraham et al. [47] observed similar rates of progression between males and females until 20 years post-diagnosis, and only women had a quicker rate of progression after this period. A prior study with a ten-year follow-up had found that females had slower initial but faster later impairment progression [48]. Thus, this point is not clear, being an important research area to understand underlying reasons for this heterogeneity in PD progression [49]. Specifically, different genetic, hormonal, neuroendocrinal, and molecular players contribute towards the differences in PD pathogenesis [12,13]. Regarding motor complications, several studies reported a high incidence in females [50,51,52]. We did not find differences in the frequency of motor fluctuations and dyskinesia at baseline between males and females. Additionally, although the UPDRS-IV score was impaired after a two-year follow-up more in females, only a trend in significance was detected. Despite this difference observed in many studies, women with PD did not receive different treatments compared with men, suggesting that non-motor fluctuations in women remain mostly undertreated [53]. In our analysis, LEDD was lower in females at baseline (all cohort) and tended to be lower in all visits (subgroup with data available in all visits), which has been previously reported [5]. However, no differences between both groups were detected when LEDD was divided by weight, which could suggest a similar management in terms of dose requirement. Interestingly, tremor was the only motor sign that decreased after the two-year follow-up in both groups in our cohort. It is well known that motor PD subtype is instable and some cases with tremor-dominant type change to non-tremor subtype in the short-term [23]. 

There is currently a large amount of evidence available on the sex differences in the spectrum of NMSs in PD patients [11]. However, there are many scales validated to assess different NMSs in PD and this could contribute to heterogeneous results in some respects [2]. PD women have more depression, sleep problems, fatigue, and pain [1,2,8,11,54,55,56,57,58,59]. However, urinary symptoms and sexual dysfunction seem to be more prevalent in men [1,2,8,58,60]. In our cohort, depression was double in women than men. Women also were receiving more frequent antidepressant agents, benzodiazepines, and analgesics. When considering NMSs as a whole, the results were inconsistent. According to Solla et al. [61], women had higher scores on the NMSS, including severe sleep difficulties, increased fatigue, and mood disorders, such as apathy, anxiety, sadness, depression, and lack of motivation, whereas men had higher sexual dysfunction levels. However, Nicoletti et al. [62] reported that the presence of NMSs was more strongly associated with male sex in a cross-sectional study conducted in 585 PD patients, but no differences were detected in another cross-sectional study conducted in 415 PD patients by Kang et al. [46]. In all these studies, the NMS burden was assessed with the same scale we used, the NMSS. Some studies suggest a greater NMS burden in males [61]. In particular, Picillo et al. [62] observed that men complained of a greater number of NMSs as compared to women in a two-year follow-up study as well. Here, we detected only a trend in significance after applying the Bonferroni test (*p* = 0.012) for a greater NMS burden increase in males after a two-year follow-up, without significant differences by sex in the change of the score of all domains of the NMSS. Although some studies reported that male PD patients have worse general cognitive abilities and that male sex is the primary predictive factor for mild cognitive impairment and its rapid progression in the severe stage of the disease [7,9,10,11,63], we did not find significant differences in the neuropsychiatric symptoms. Regarding ICD and/or CB, the frequency was according to the literature [64] and hypersexuality was the only ICD significantly more frequent in males, being present in 7.6% of them. A recent review reported a mean prevalence of hypersexuality of 3.5% in PD patients and quite convincing demographic data indicating that patients are often males on dopamine agonists [65]. Finally, females have significantly less social support, more psychological distress, and worse self-reported disability and health-related QoL compared to males [47,66,67,68]. In our analysis, we detected differences with a worse perception by females, especially in emotional well-being, pain, and discomfort. Few studies have analyzed differences in the changes over time to develop the ADL in PD between sexes. We detected an impairment in the functional capacity in both sexes after the two-year follow-up conducted, but more significantly in women, as Sperens et al. [69] reported in an eight-year follow-up study conducted in 129 PD patients, with a worse impairment in women in domains such as shopping and cleaning. Other studies emphasize the importance of the progressive disability of women with PD, influencing aspects, such as polypharmacy or comorbidities, that translate into a consumption of health resources [56].

The present study has some limitations. Our findings may not be applicable to all PD patients in the community clinical setting because PD patients enrolled in the COPPADIS study represent a selected population with less disability at baseline than the general population (e.g., no older than 75 years old, not being under a second-line therapy, etc.). For some variables, the information was not collected in all cases and the data in the follow-up were obtained in 496 patients of 681 initially included in the baseline analysis (72.8%), so they could influence the results. However, this percentage is even higher in other longitudinal prospective studies [70] and a bias of withdrawals of more affected patients was excluded after comparing subjects who were lost to follow-up vs. those who were not. Another important aspect is that, due to the large sample size, some differences observed could be statistically significant but not a minimum clinically important difference (e.g., differences detected in motor aspects such as hypomimia, speech, and rigidity) [71], so it is necessary to be cautious about the relevance of some differences detected. Moreover, the observational nature of the study does not provide support for a cause–effect relationship but can only suggest correlation between variables. Furthermore, the medication dose was not adjusted for the body mass index because these data were not collected, but we did do so by weight. On the other hand, and with the intention to reduce the instance of a false positive, we applied the Bonferroni test with a significant *p* value definition of ≤ 0.002. Finally, we described the differences between men and women with PD in our cohort, but we did not analyze other markers in this manuscript that could explain the reason for them [72]. Nonetheless, the strengths of our study include a large sample size, a very thorough assessment, a prospective longitudinal follow-up design, and the extensive clinical and demographic information recorded.

In conclusion, the present study supports the idea that there are sex differences in PD. Symptoms such as depression, fatigue, or pain seem to be more frequent and/or severe in females, whereas others such as hypomimia, speech problems, rigidity, or hypersexuality were more common in males. Moreover, women could have a worse perception of their QoL. All in all, more studies are needed to better understand the differences between males and females with PD, especially in the long-term follow-up, and their causes.

## Figures and Tables

**Figure 1 jcm-12-01329-f001:**
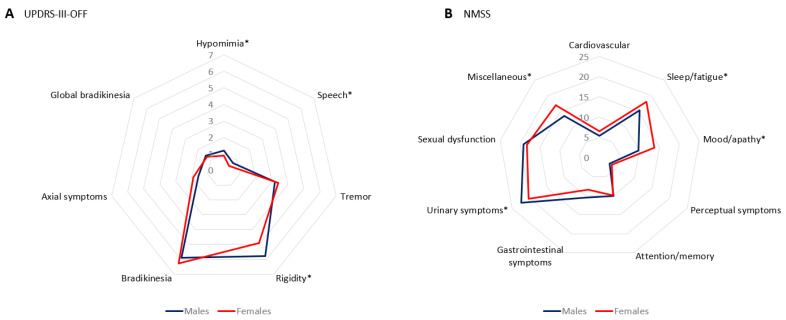
(**A**) Mean score of sub-scores of the UPDRS-III-OFF at baseline in PD males vs. females; *, *p* < 0.0001. (**B**) Mean score on each domain of the NMSS at baseline in PD males vs. females; *, *p* < 0.05; Sleep/fatigue, *p* = 0.041; Mood/apathy, *p* = 0.003; Urinary symptoms, *p* = 0.047; Miscellaneous, *p* = 0.010.

**Figure 2 jcm-12-01329-f002:**
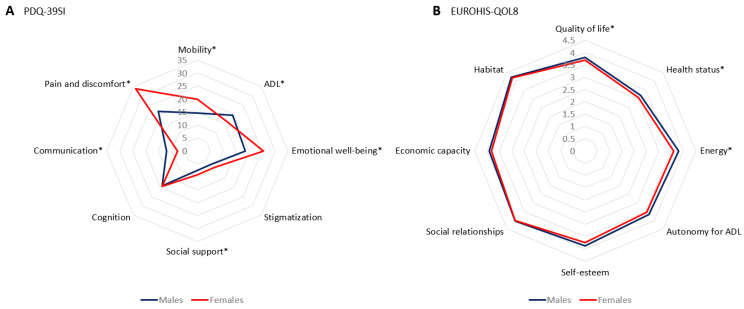
(**A**) Mean score of each domain of the PDQ-39 at baseline in PD males vs. females; *, *p* < 0.05; Mobility, *p* = 0.01; ADL, Activities of daily living, *p* = 0.034; Emotional well-being, *p* < 0.0001; Social support, *p* = 0.046; Communication, *p* < 0.0001; Pain and discomfort, *p* < 0.0001. ADL: Activities of daily living (ADL): (**B**) Mean score on each domain of the EUROHIS-QOL8 at baseline in PD males vs. females; *, *p* < 0.05; Quality of life, *p* = 0.014; Health status, *p* = 0.020; Energy, *p* = 0.04.

**Figure 3 jcm-12-01329-f003:**
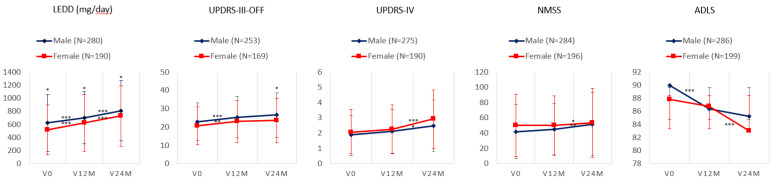
Mean and standard deviation at V0, V12M, and V24M in males vs. females in the LEDD (mg/day) and the score on UPDRS-III-OFF, UPDRS-IV, NMSS, and ADLS; *, *p* < 0.05; **, *p* < 0.001; ***, *p* < 0.0001. The symbol above the line represents the significance of the change between one visit and another (in men and women) while the symbol above the point/diamond represents the difference in that visit between men and women. ADLS, Schwab and England Activities of Daily Living Scale; LEDD, levodopa equivalent daily dose; NMSS, Non-Motor Symptoms Scale; UPDRS, Unified Parkinson’s Disease Rating Scale.

**Table 1 jcm-12-01329-t001:** Sociodemographic and lifestyle variables and comorbidities in PD patients from the COPPADIS cohort at baseline (V0) regarding sex (*n* = 681).

	All Sample (*n* = 681)	Males (*n* = 410)	Females (*n* = 271)	*p*
Age	62.54 ± 8.93	62.36 ± 8.73	62.8 ± 9.24	0.297
Education level (%):				0.006
-Primary	42.1	38	48.1	
-Secondary	31.5	31.5	31.5	
-University	26.5	30.5	20.4	
Civil status (%):				**<0.0001**
-Married	78.4	83.9	70	
-Single	7.1	4.4	11.1	
-Widowed	6.8	3.7	11.5	
-Divorced	6.6	6.8	6.3	
-Other	1.1	1.2	1.1	
Living style (%):				**<0.0001**
-With the partner	79.3	85.1	70.4	
-Alone	10.7	7.6	15.6	
-With a son/daughter	5.3	2.7	9.3	
-Other	4.7	4.6	4.8	
Habitat (%):				0.438
-Rural (<5.000)	11.8	12.7	10.4	
-Semiurban (5.000–20.000)	18.9	17.6	43.8	
-Urban (>20.000)	69.4	69.8	68.8	
Arterial hypertension (%)	33.7	35.5	31	0.229
Diabetes mellitus (%)	9.1	10.3	7.4	0.200
Dyslipidemia (%)	30.1	30.1	30.3	0.959
Cardiopathy (%)	7.9	10	4.8	0.014
Cardiac arrhythmia (%)	5.1	5.9	4.1	0.296
Smoking (%):				**<0.0001**
-Smoker	9.4	9.5	9.2	
-Ex-smoker	30	35.9	21	
-Non-smoker	60.6	54.5	69.8	
Alcohol consumption (%):				**<0.0001**
-Excessive drinker	0.3	0.5	0	
-Non-excessive drinker	20.6	28.4	8.9	
-Non-drinker	79.1	71.1	91.1	
To be receiving (%):				
-Antidepressant agent	24.3	18.1	33.3	**<0.0001**
-Benzodiazepine	16	11.2	23.2	**<0.0001**
-Antipsychotic	2.2	3.2	0.7	0.034
-Analgesic	24.3	19.8	31	**0.002**
Number of non-antiparkinsonian drugs	2.58 ± 2.46	2.4 ± 2.36	2.86 ± 2.6	0.029
Assistance to a patient association (%)	14.1	12	17.4	0.047
Speech therapy (%)	11.3	11.5	11.2	0.925
Physiotherapy (%)	28	27.6	28.5	0.800
Regular exercise (%)	69.8	72.6	65.6	0.050
Cognitive stimulation therapy (%)	15.8	15.2	16.7	0.598

Chi-square and Mann–Whitney–Wilcoxon test were used. The results represent mean ± SD or %. A *p* ≤ 0.002 was considered significant (Bonferroni correction).

**Table 2 jcm-12-01329-t002:** Different PD-related variables in PD patients from the COPPADIS cohort at baseline (V0) regarding sex (*n* = 681).

	All Sample (*n* = 681)	Males (*n* = 410)	Females (*n* = 271)	*p*
**Disease Treatments**				
Years from symptoms onset	5.48 ± 4.38	5.66 ± 4.65	5.21 ± 4.11	0.259
To be receiving (%):				
-Levodopa	72.2	73.5	70.1	0.337
-DA	68.8	72.1	63.8	0.024
-MAO-B inhibitor	73.5	77.2	67.9	0.007
-COMT inhibitor	18.6	20.3	16	0.160
-Amantadine	7.8	7.1	9	0.382
Number of antiparkinsonian drugs	2.43 ± 1.11	2.32 ± 1.09	2.51 ± 1.11	0.011
Daily dose of levodopa (mg/day)	346.13 ± 309.64	365.95 ± 319.73	316.44 ± 291.96	0.047
DA equivalent daily dose (mg/day)	167.49 ± 155.08	181.06 ± 159.41	147.17 ± 146.32	0.005
LEDD (mg/day)	566.13 ± 515.75	607.79 ± 432.61	503.71 ± 381.49	**0.002**
Weight (kgs)	75.77 ± 13.77	80.89 ± 11.84	68.04 ± 12.31	**<0.0001**
LEDD/Kgs	7.64 ± 5.8	7.61 ± 5.5	6.69 ± 7.17	0.676
**Motor Symptoms**				
HY stage (OFF)	2 [2,2]	2 [2,2]	2 [1.5,2]	0.810
-Stage from 3 to 5 (%)	9.5	8.9	10.4	0.521
UPDRS-III (OFF)	22.8 ± 11.21	23.24 ± 11.28	22.15 ± 11.1	0.265
UPDRS-IV	2.01 ± 2.41	1.94 ± 2.82	2.11 ± 2.58	0.994
-Motor fluctuations (%)	32.8	32.5	33.3	0.825
-Dyskinesia (%)	18.7	17.9	19.8	0.552
FOGQ	3.79 ± 4.59	3.79 ± 4.56	3.79 ± 4.58	0.994
-FOG (%)	34.5	35.1	33.6	0.680
-Falls (%)	13.1	10.6	16.7	0.022
Tremoric motor phenotype (%)	54.9	57.7	50.6	0.067
**Non-Motor Symptoms**				
PD-CRS	91.27 ± 15.83	91.24 ± 15.59	91.32 ± 16.21	0.974
-Cognitive impairment (%)	25.4	25.5	25.3	0.950
NMSS	45.26 ± 37.99	43.49 ± 36.89	47.94 ± 39.52	0.208
-Very severe NMS burden (%)	19.5	18.2	21.5	0.289
BDI-II	8.75 ± 7.34	7.95 ± 6.84	9.95 ± 7.89	**0.001**
-Major depression (%)	16.3	12	22.9	**<0.0001**
PDSS	114.84 ± 26.99	115.27 ± 26.95	115.27 ± 26.95	0.408
QUIP-RS	4.33 ± 8.29	4.67 ± 8.56	3.83 ± 7.85	0.049
-ICD and/or CB (%)	17	17.4	16.3	0.703
NPI	6.06 ± 8.88	5.54 ± 7.23	6.86 ± 8.88	0.362
VAS-PAIN	2.62 ± 2.92	2.13 ± 2.62	3.37 ± 3.17	**<0.0001**
-Pain (%)	57	51.7	64.9	**0.001**
VAFS–physical	2.95 ± 2.75	2.55 ± 2.58	3.57 ± 2.88	**<0.0001**
VAFS–mental	2.16 ± 2.55	1.91 ± 2.34	2.54 ± 2.8	0.012
**Autonomy and QOL**				
PDQ-39SI	17.18 ± 13.56	16.05 ± 13.33	18.88 ± 13.76	**0.002**
EUROHIS-QOL8	3.78 ± 0.55	3.82 ± 0.53	3.71 ± 0.56	0.009
ADLS	88.56 ± 10.45	88.76 ± 10.09	88.26 ± 10.98	0.835
-Dependency for ADL (%)	9	7.6	11.1	0.113

Chi-square and Mann–Whitney–Wilcoxon test were used. The results represent mean ± SD, median [p25, p75] or %. The value was not available on all variables, with the smallest *n* being 574 for the NPI. A *p* ≤ 0.002 was considered significant (Bonferroni correction). It is shown in bold. ADL, activities of daily living; ADLS, Schwab and England Activities of Daily Living Scale; BDI-II, Beck Depression Inventory-II; DA, dopamine agonist; MAO-B, monoamine oxidase-B; COMT, catechol-O-methyltransferase; FOG, freezing of gait; FOGQ, Freezing Of Gait Questionnaire; LEDD, levodopa equivalent daily dose; HY, Hoenh and Yahr; NMSS, Non-Motor Symptoms Scale; NPI, Neuropsychiatric Inventory; PD-CRS, Parkinson’s Disease Cognitive Rating Scale; PDSS, Parkinson’s Disease Sleep Scale; VAS-Pain, Visual Analog Scale-Pain; VAFS; Visual Analog Fatigue Scale; QOL, quality of life; PDQ, Parkinson’s disease Questionnaire; QUIP-RS, Questionnaire for Impulsive-Compulsive Disorders in Parkinson’s Disease-Rating Scale; ICD, impulse control disorder; CB, compulsive behavior; UPDRS, Unified Parkinson’s Disease Rating Scale.

**Table 3 jcm-12-01329-t003:** Differences in motor aspects, cognition, impulsive-compulsive behaviors, and QoL between men and women with PD from the COPPADIS cohort at baseline (V0) (*n* = 681).

	All Sample (*n* = 681)	Males (*n* = 410)	Females (*n* = 271)	*p*
**Motor Symptoms**				
**UPDRS-III-Off**	22.8 ± 11.21	23.24 ± 11.28	22.15 ± 11.1	0.265
Hypomimia	1.16 ± 0.75	1.29 ± 0.76	0.96 ± 0.69	**<0.0001**
Speech	0.64 ± 0.68	0.78 ± 0.71	0.43 ± 0.64	**<0.0001**
Tremor	3.29 ± 3.15	3.2 ± 3.19	3.43 ± 3.09	0.226
Rigidity	5.5 ± 3.26	5.87 ± 3.3	4.95 ± 3.12	**<0.0001**
Bradykinesia	6.11 ± 3.78	5.96 ± 3.7	6.32 ± 3.9	0.324
Axial symptoms	1.77 ± 1.86	1.67 ± 1.74	1.93 ± 2.04	0.613
Global bradykinesia	1.35 ± 0.9	1.36 ± 0.88	1.33 ± 0.92	0.777
**Non-Motor Symptoms**				
**PD-CRS total score**	91.27 ± 15.83	91.24 ± 15.59	91.32 ± 16.21	0.974
**PD-CRS FS sub-score**	63.67 ± 14.49	63.5 ± 14.52	63.93 ± 14.47	0.794
Immediate verbal memory	8.07 ± 2.1	7.91 ± 2.13	8.32 ± 2.03	0.007
Sustained attention	8.47 ± 1.89	8.56 ± 1.81	8.33 ± 2	0.160
Working memory	6.85 ± 2.4	7.07 ± 2.38	6.51 ± 2.38	0.003
Clock drawing	8.94 ± 1.69	8.96 ± 1.77	8.92 ± 1.57	0.499
Delayed verbal memory	5.54 ± 2.72	5.36 ± 2.64	5.81 ± 2.82	0.049
Alternating verbal fluency	11.25 ± 4.74	11.26 ± 4.4	11.25 ± 4.58	0.723
Action verbal fluency	14.59 ± 5.69	14.44 ± 5.85	14.8 ± 5.43	0.266
**PD-CRS PC sub-score**	27.6 ± 3.37	27.74 ± 3.31	27.39 ± 3.46	0.083
Confrontation naming	18.05 ± 3.06	18.23 ± 2.96	17.79 ± 3.19	0.012
Clock copy	9.55 ± 1.21	9.51 ± 1.29	9.6 ± 1.05	0.712
**NMSS**	45.26 ± 37.99	43.49 ± 36.89	47.94 ± 39.52	0.208
Cardiovascular	5.85 ± 10.21	5.41 ± 10.1	6.52 ± 10.36	0.137
Sleep/fatigue	16.42 ± 16.12	15.35 ± 15.63	18.03 ± 16.72	0.041
Mood/apathy	11.38 ± 16.45	9.79 ± 15.19	13.78 ± 17.96	0.003
Perceptual symptoms	3.13 ± 8.71	2.86 ± 7.98	3.52 ± 9.71	0.529
Attention/memory	9.97 ± 14.05	10 ± 14.13	9.92 ± 13.96	0.912
Gastrointestinal symptoms	9.69 ± 13.13	10.51 ± 13.9	8.46 ± 11.79	0.057
Urinary symptoms	21.61 ± 22.48	22.48 ± 22.22	20.29 ± 22.84	0.047
Sexual dysfunction	18.79 ± 25.69	19.1 ± 25.31	18.32 ± 26.3	0.201
Miscellaneous	14.92 ± 15.49	13.56 ± 14.49	16.98 ± 16.71	0.010
**QUIP-RS**				
Any ICD and/or CB (%)	17	17.4	16.3	0.703
Any ICD (%)	12.2	12.5	11.8	0.782
Compulsive gambling (%)	1.5	1.9	0.8	0.268
Hypersexuality (%)	4.7	7.6	0.4	**<0.0001**
Compulsive shopping (%)	2.4	1.6	3.7	0.112
Compulsive eating (%)	6.7	5.2	8.9	0.067
Any CB (%)	9.3	9.3	9.3	0.972
Hobbyism–punding (%)	6.9	6.8	6.9	0.962
Compulsive medication (%)	3.8	3.8	3.7	0.921
**Quality of Life**				
**PDQ-39SI**	17.18 ± 13.56	16.05 ± 13.33	18.88 ± 13.76	**0.002**
Mobility	16.69 ± 19.31	14.59 ± 17.88	19.87 ± 20.92	**0.001**
Activities of daily living	18.1 ± 18.71	19.41 ± 19.69	16.11 ± 16.98	0.034
Emotional well-being	21.4 ± 19.93	18.6 ± 18.72	25.64 ± 20.97	**<0.0001**
Stigmatization	13.62 ± 19.57	13.1 ± 18.95	14.4 ± 20.49	0.665
Social support	8.21 ± 16.55	7.57 ± 16.88	9.18 ± 16.02	0.046
Cognition	19.22 ± 17.81	19.14 ± 17.51	19.33 ± 18.3	0.939
Communication	10.3 ± 15.34	11.99 ± 16.24	7.74 ± 13.52	**<0.0001**
Pain and discomfort	26.36 ± 22.72	21.46 ± 20.2	33.76 ± 24.31	**<0.0001**
**EUROHIS-QOL8**	3.78 ± 0.55	3.82 ± 0.53	3.71 ± 0.56	0.009
Quality of life	3.8 ± 0.71	3.86 ± 0.7	3.73 ± 0.72	0.014
Health status	3.16 ± 0.88	3.23 ± 0.87	3.07 ± 0.87	0.020
Energy	3.76 ± 0.82	3.84 ± 0.77	3.65 ± 0.88	0.004
Autonomy for ADL	3.62 ± 0.86	3.66 ± 0.84	3.55 ± 0.89	0.062
Self-esteem	3.81 ± 0.81	3.87 ± 0.79	3.73 ± 0.84	0.054
Social relationships	4.04 ± 0.71	4.04 ± 0.72	4.03 ± 0.7	0.798
Economic capacity	3.86 ± 0.77	3.9 ± 0.78	3.81 ± 0.76	0.095
Habitat	4.23 ± 0.68	4.25 ± 0.68	4.21 ± 0.69	0.383

Chi-square and Mann–Whitney–Wilcoxon test were used. The results represent mean ± SD, median [p25, p75] or %. The value was not available on all variables, with the smallest *n* being 622 for the UPDRS-III. A *p* ≤ 0.002 was considered significant (Bonferroni correction). CB, compulsive behavior; ICD, impulse control disorder; NMSS, Non-Motor Symptoms Scale; PD-CRS, Parkinson’s Disease Cognitive Rating Scale; FS; fronto-subcortical; PC, posterior-cortical; QUIP-RS, Questionnaire for Impulsive-Compulsive Disorders in Parkinson’s Disease-Rating Scale; UPDRS, Unified Parkinson’s Disease Rating Scale.

**Table 4 jcm-12-01329-t004:** Changes from the baseline visit (V0) to the two-year follow-up visit (V24M) in different PD-related variables in PD patients from the COPPADIS cohort regarding sex (*n* = 496).

	Males V0	Males V24M	Cohen’s	*p* ^a^	Females V0	Females V24M	Cohen’s	*p* ^b^	*p* ^c^	*p* ^d^
	*n* = 294	*n* = 294	Test		*n* = 202	*n* = 202	Test			
**Dose therapy**										
LEDD (mg/day)	625.25 ± 428.6	803.88 ± 455	0.73	**<0.0001**	509.48 ± 381.37	719.22 ± 462.01	0.96	**<0.0001**	0.249	0.007
LEDD/weight	7.61 ± 5.54	10.82 ± 6.07	0.87	0.131	7.69 ± 6.17	10.53 ± 7.51	0.71	0.005	0.859	0.706
**Motor Symptoms**										
UPDRS-III (OFF)	22.84 ± 10.56	26.3 ± 12.01	0.56	**<0.0001**	20.74 ± 10.38	23.76 ± 12.42	0.41	**<0.0001**	0.686	0.554
Hypomimia	1.24 ± 0.72	1.35 ± 0.71	0.22	0.014	0.93 ± 0.66	0.99 ± 0.73	0.15	0.254	0.539	**<0.0001**
Speech	0.72 ± 0.72	0.81 ± 0.75	0.13	0.032	0.4 ± 0.58	0.46 ± 0.64	0.04	0.084	0.808	**<0.0001**
Tremor	3.3 ± 3.18	2.36 ± 2.62	−0.47	**<0.0001**	3.26 ± 0.21	2.41 ± 0.19	−0.34	**<0.0001**	0.889	0.763
Rigidity	5.83 ± 3.21	6.1 ± 3.62	0.17	0.131	4.76 ± 2.99	5.22 ± 3.19	0.18	0.030	0.514	0.018
Bradykinesia	5.86 ± 3.6	7.3 ± 4.19	0.55	**<0.0001**	6.16 ± 3.76	7.02 ± 4.37	0.29	0.001	0.253	0.226
Axial symptoms	1.67 ± 1.76	3.03 ± 2.19	0.72	**<0.0001**	2 ± 2.01	3.16 ± 2.31	0.41	**<0.0001**	0.462	0.237
Global bradykinesia	1.37 ± 0.89	1.6 ± 0.93	0.39	**<0.0001**	1.32 ± 0.92	1.46 ± 1.05	0.19	0.034	0.424	0.898
UPDRS-IV	1.91 ± 2.25	2.46 ± 2.64	0.31	**<0.0001**	2.08 ± 2.6	2.94 ± 2.89	0.47	**<0.0001**	0.164	0.105
FOGQ	3.65 ± 4.61	4.83 ± 5.17	0.43	**<0.0001**	3.95 ± 4.85	4.97 ± 5.16	0.34	0.001	0.376	0.014
**Non-Motor Symptoms**										
PD-CRS	91.61 ± 15.55	89.89 ± 17.22	−0.21	0.011	92.4 ± 15.9	90.92 ± 19.35	−0.15	0.068	0.886	0.615
PD-CRS FS sub-score	63.84 ± 14.29	61.99 ± 15.88	−0.22	0.003	64.75 ± 14.44	63.2 ± 17.24	−0.12	0.029	0.864	0.237
PD-CRS PC sub-score	27.98 ± 3.14	27.97 ± 2.9	−0.03	0.950	27.55 ± 3.35	27.25 ± 3.5	−0.11	0.200	0.449	0.021
NMSS	42.44 ± 35.81	52.25 ± 41.45	0.43	**<0.0001**	48.99 ± 40.08	55.59 ± 43.88	0.25	0.013	0.469	0.012
-Cardiovascular	4.65 ± 8.82	11.73 ± 14.86	0.71	**<0.0001**	6.58 ± 11.1	12.06 ± 13.26	0.57	**<0.0001**	0.300	0.207
-Sleep/fatigue	14.71 ± 14.98	18.8 ± 17.84	0.33	**<0.0001**	18.29 ± 16.65	19.83 ± 17.23	0.11	0.212	0.086	0.028
-Mood/apathy	9.2 ± 14.3	11.2 ± 16.77	0.33	0.015	14.2 ± 17.97	16 ± 19.49	0.1	0.159	0.934	**<0.0001**
-Perceptual symptoms	2.98 ± 8.96	5.55 ± 13.29	0.37	**<0.0001**	3.94 ± 10.39	5.43 ± 11.72	0.17	0.082	0.275	0.466
-Attention/memory	10.1 ± 14.84	12.59 ± 16.81	0.27	0.004	10.15 ± 14.65	12.68 ± 17.72	0.22	0.039	0.971	0.723
-Gastrointestinal	10.11 ± 13.76	11.82 ± 13.99	0.24	0.018	9.02 ± 12.51	12.67 ± 14.88	0.38	**<0.0001**	0.123	0.984
-Urinary symptoms	22.35 ± 21.79	22.63 ± 22.44	0.02	0.109	21.02 ± 23.71	21.94 ± 23.36	0.11	0.517	0.237	0.493
-Sexual dysfunction	18.64 ± 24.96	24.77 ± 28.5	0.31	**<0.0001**	19.04 ± 26.98	19.25 ± 26.27	0.02	0.821	0.101	0.219
-Miscellaneous	13.01 ± 14.06	14.39 ± 14.11	0.19	0.083	17.45 ± 17.12	18.77 ± 16.78	0.11	0.305	0.921	**<0.0001**
BDI-II	7.34 ± 6.47	8.23 ± 7.23	0.16	0.043	9.52 ± 8.48	9.02 ± 7.92	−0.08	0.417	0.062	0.003
PDSS	117.78 ± 24.3	117.71 ± 25.86	−0.04	0.962	115.89 ± 25.13	117.97 ± 24.09	0.11	0.260	0.553	0.160
QUIP-RS	4.98 ± 9.21	5.12 ± 9.59	0.03	0.815	3.57 ± 6.98	3.63 ± 8.29	0.02	0.912	0.954	0.187
NPI	5.39 ± 7.52	5.51± 9.01	0.02	0.839	6.67 ± 8.64	7.27 ± 10.12	0.07	0.406	0.492	0.014
VAS-PAIN	2.11 ± 2.68	2.32 ± 2.59	0.12	0.240	3.31 ± 3.15	3.78 ± 2.99	0.15	0.079	0.398	**<0.0001**
VAFS–physical	2.51 ± 2.54	2.77 ± 2.65	0.15	0.137	3.29 ± 2.82	3.68 ± 2.94	0.13	0.088	0.551	**<0.0001**
VAFS–mental	1.86 ± 2.29	2.01 ± 2.45	0.11	0.353	2.39 ± 2.76	2.37 ± 2.79	0	0.937	0.005	N. A.
**Autonomy and QOL**										
PDQ-39SI	15.37 ± 12.75	17.96 ± 15.45	0.32	**<0.0001**	18.49 ± 13.45	23.27 ± 17.24	0.53	**<0.0001**	0.027	N. A.
EUROHIS-QOL8	3.82 ± 0.49	3.76 ± 0.56	−0.13	0.063	3.69 ± 0.57	3.72 ± 0.61	0.07	0.046	0.040	N. A.
ADLS	89.12 ± 9.08	85.23 ± 12.58	0.49	**<0.0001**	87.91 ± 11.42	82.76 ± 14.69	0.57	**<0.0001**	0.176	0.001

*p* values were computed using general linear models (GLM) repeated measures. The results represent mean ± SD. The value was not available on all variables, with the smallest *n* being 387 for the UPDRS-III; *p*
^a^, change over time (V24M vs. V0) in males; *p*
^b^, change over time (V24M vs. V0) in females. Age at V0, and LEDD (levodopa equivalent daily dose) (except for assessing changes in this variable) at V0 and at V24M were included as covariates; *p*
^c^, group visit interaction; *p*
^d^, males vs. females. Males vs. females is not applicable if test of interaction was significant (a significant test of interaction means the rates of changes over time are different between the two groups). All patients with the data at V0 and V24M were included for each comparative analysis. A *p* < 0.002 (in bold) was considered significant (Bonferroni correction). ADLS, Schwab and England Activities of Daily Living Scale; BDI-II, Beck Depression Inventory-II; FOGQ, Freezing Of Gait Questionnaire; LEDD, levodopa equivalent daily dose; NMSS, Non-Motor Symptoms Scale; NPI, Neuropsychiatric Inventory; PD-CRS, Parkinson’s Disease Cognitive Rating Scale; FS, fronto-subcortical; PC, posterior-cortical; PDSS, Parkinson’s Disease Sleep Scale; QUIP-RS, Questionnaire for Impulsive-Compulsive Disorders in Parkinson’s Disease-Rating Scale; UPDRS, Unified Parkinson’s Disease Rating Scale; VAFS, Visual Analog Fatigue Scale; QoL, quality of life; PDQ.39, the 39-item Parkinson’s disease Questionnaire; VAS-Pain, Visual Analog Scale-Pain.

## Data Availability

The protocol and the statistical analysis plan are available on request. Deidentified participant data are not available for legal and ethical reasons.

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
