# Peer review of "Sex Differences in Motor and Non-Motor Symptoms among Spanish Patients with Parkinson’s Disease"

_jcm, 2023, doi:10.3390/jcm12041329_

Round 1

Reviewer 1 Report

This is a very interesting work by Garcia-Santos et al. that studies sex differences in the prevalence and progression of non-motor symptoms in a Spanish population. I would like to congratulate the authors for their comprehensive work and for addressing important and relevant aspects of sex differences in PD, such as differences in PD-related drugs other than levodopa and differences in the prescription of treatments for non-motor symptoms, like depression of anxiety. This is an understudied topic, and deserves to be incorporated in clinical studies, like this one. Also, authors have studied differences in habits, which might interfere with disease progression and symptomatology, so this also deserves to be highlighted. Finally, nowadays there are only few studies that have analyzed the natural evolution of PD symptomatology (including motor and non-motor symptoms), so this work is very relevant in the field.

Although the contribution of the authors is clearly outstanding, I feel that some points need to be clarified in the current work:

1-. Why was the relative change in NMSS and PDQ-39 subscores used instead of the absolute values (line 133-134)? Which is the rationale behind this? In Table 4, absolute values are provided for each NMSS domain, which might be misleading if data has been analyzed otherwise.

2.- I really believe that using linear mixed-effect models (LMM) would be more appropriate for current analyses. LMM can handle missing data (unlike repeated measures ANOVA), and avoids unnecessarily high number of statistical comparisons (such as in Fig.3) that increases the number of false-positive findings. Also, I do not really understand why age at baseline and age at follow-up was introduced in the model (line265), as well as LEDD at both time points (this makes sense, but not sure that this is the best approximation). Usually, in LMM, time-varying covariates can be introduced with an interaction factor with time factor.

3.- If not willing to re-analyze the data with LMM, because GLM is also well-suited (although not ideal from my point of view), I would at least suggest to make the following changes or to clarify the following points:

3.1.-p-value (c) in Table 4. Females vs. males. Does this p-value correspond to “mean” values of females and males? That is, does that p-value correspond to the pooled values of V0 and V24? Or it is just the baseline comparison between males and females?

3.2.- This is related to the previous comment. I think that in the statistical analyses it should be clarified how the GLM was constructed. Which were the outcome variables (I guess it was the absolute difference between V0 and V24), and which were the predictors (sex?) and covariables (age at baseline? LEDD at V0 and V24?). A sentence describing the GLM would let other authors to reproduce their analyses.

3.3.- There is a considerable missing of data at follow-up (FU). Concretely, 420 PD males were analyzed at baseline and 294 at FU (116 PD males lost to follow-up, 28.3%). 271 PD females at baseline and 202 at FU (69 dropouts, 25.5%). This should be stated in results. Lost to follow-up is a problem in clinical research that it is difficult to handle. However, I think it would be great to know if patients that were lost at FU were different at baseline from those who were not, and write this shortly in results.

3.4.- If authors used age at baseline and age at FU in GLM, I would suggest them to reconsider using only age at baseline in the analyses (I am not sure from the text if one or the other was used). If only age at baseline was used, write it accordingly in lines 163, line 265…

Some minor comments:

4.- In Tables, it would be clearer if the first column was center at left and using a different indentation for titles and subtitles.

5.- As there is a lot of information in Tables, it would help a lot if authors could highlight somehow the significant p-valued (in bold or using asterisks after p-values, for example).

6.- Line 131, “except in” seems more appropriate that “apart from”

7.- Standard deviations should be incorporated in Figure 3.

8.- Line 254, “Collectively, the NMS burden (NMSS total score) impaired after the 2-year follow-up in both groups”  I would suggest to change it for “collectively, we observed an impairment of NMS burden (NMSS total score) after…”

9.- Discussion. The authors state that LEDD is lower in women, but it has been demonstrated that this is not true after correcting for BMI. If the field is moving towards personalized medicine and sex differences should be considered, the drug posology should also be adapted to BMI.

10.-Results and discussion. According to Table 4 , there are not differences in the progression of NMS between males and females. This is a very interesting finding that it is not discussed in the text.

11.- As there is a dropout rate higher than 20%, a sentences stating how missing data could influence the current results is worthy.

Author Response

Reviewer 1

This is a very interesting work by Garcia-Santos et al. that studies sex differences in the prevalence and progression of non-motor symptoms in a Spanish population. I would like to congratulate the authors for their comprehensive work and for addressing important and relevant aspects of sex differences in PD, such as differences in PD-related drugs other than levodopa and differences in the prescription of treatments for non-motor symptoms, like depression of anxiety. This is an understudied topic, and deserves to be incorporated in clinical studies, like this one. Also, authors have studied differences in habits, which might interfere with disease progression and symptomatology, so this also deserves to be highlighted. Finally, nowadays there are only few studies that have analyzed the natural evolution of PD symptomatology (including motor and non-motor symptoms), so this work is very relevant in the field.

Although the contribution of the authors is clearly outstanding, I feel that some points need to be clarified in the current work:

1-. Why was the relative change in NMSS and PDQ-39 subscores used instead of the absolute values (line 133-134)? Which is the rationale behind this? In Table 4, absolute values are provided for each NMSS domain, which might be misleading if data has been analyzed otherwise.

AUTHORS – Thank you for your comment. In Methods, it is explained that “Total score of the PDQ-39 (PDQ-39SI) and each domain of the NMSS and the PDQ-39 was expressed as a percentage: (score/total score) x 100 [20]”. It is frequent to show results in this way in the literature, including in many other publications of the COPPADIS cohort, with the intention of being able to compare and express each domain in a figure (e.g., 1B, 2A) given that the maximum total value of each one is different (for example, a maximum total score of 24 in the domain 1 of the NMSS vs. 72 in domain 3). However and by mistake, in Table 3 and Figure 2B, absolute values were used for the NMSS-domain 1 (in the database one variable appears after the other)). It has been corrected and expressed as a percentage. Please, note that sample size is not exactly the same for Table 3 and 4. Many thanks again for your important observation.

2.- I really believe that using linear mixed-effect models (LMM) would be more appropriate for current analyses. LMM can handle missing data (unlike repeated measures ANOVA), and avoids unnecessarily high number of statistical comparisons (such as in Fig.3) that increases the number of false-positive findings. Also, I do not really understand why age at baseline and age at follow-up was introduced in the model (line265), as well as LEDD at both time points (this makes sense, but not sure that this is the best approximation). Usually, in LMM, time-varying covariates can be introduced with an interaction factor with time factor.

AUTHORS – Thank you for your comment. GLM repeated measure was used to test whether the mean differences of the score of different scales (motor; NMS; QoL; autonomy for ADL) between visits (V0, V12M, and/or V24M) were significant. The GLM repeated measures procedure provides analysis of variance when the same measurement is made several times on each subject or case. We have used previously GLM repeated measure in many publications from the COPPADIS cohort (Santos-García D et al. J Pers Med 2021;11:326; Santos-García et al. J Parkinsons Dis 2022;12:315-31; Santos-García et al. Diagnostics 2021;11:1801; Santos-García et al. NPJ Parkinsons Dis 2021;7:118; Santos-García et al. J Parkinson Dis 2022;12:935-55; Santos-García et al. Diagnostics 2022;12:1147; Santos-García et al. J Clin Neurol 2023 [epub ahead of print]). In all models, an interaction for visit and group was tested before testing for a group difference over time. Other groups have used GLM analyzing prospective longitudinal data such as from the PPMI (e.g., Simuni T, Caspell-Garcia C, Coffey CS, et al. Baseline prevalence and longitudinal evolution of non-motor symptoms in early Parkinson's disease: the PPMI cohort. J Neurol Neurosurg Psychiatry 2018;89:78-88). Only age at baseline was included in the model but LEDD was included at both times, at V0 and at V24M. We rephrase the sentence: “Age at baseline, and levodopa equivalent daily dose (LEDD [32]) at baseline and at V24M were included as covariates”.

3.- If not willing to re-analyze the data with LMM, because GLM is also well-suited (although not ideal from my point of view), I would at least suggest to make the following changes or to clarify the following points:

3.1.-p-value (c) in Table 4. Females vs. males. Does this p-value correspond to “mean” values of females and males? That is, does that p-value correspond to the pooled values of V0 and V24? Or it is just the baseline comparison between males and females?

AUTHORS – Thank you for your comment. It is explained in the table foot legend: pc, group visit interaction; pd, males vs females. Males vs females is not applicable if test of interaction was significant (a significant test of interaction means the rates of changes over time are different between the two groups). All patients with the data at V0 and V24M were included for each comparative analysis

3.2.- This is related to the previous comment. I think that in the statistical analyses it should be clarified how the GLM was constructed. Which were the outcome variables (I guess it was the absolute difference between V0 and V24), and which were the predictors (sex?) and covariables (age at baseline? LEDD at V0 and V24?). A sentence describing the GLM would let other authors to reproduce their analyses.

AUTHORS – Thank you for your comment. It has been described in Methods: “General linear model (GLM) repeated measure were used to test for changes in various scores (motor; NMS; QoL; autonomy for ADL) over time (V24M vs V0 and/or V24M vs V12M and V12M vs V0) separately in each group (men vs women) and test for differences between groups over time. Age at baseline, and levodopa equivalent daily dose (LEDD [32]) at baseline and at V24M were included as covariates. In the latter models, an interaction for visit and group was tested before testing for a group difference over time. Cohen´s d formula was applied for measuring the effect size in both groups, males and females. It was considered: < 0.2 – Negligible; 0.2 – 0.49 – Small; 0.50 – 0.79 – Moderate; ≥ 0.80 – Large. The Bonferroni method was used to p value correction for multiple comparisons and a p ≤ 0.002 was considered significant”.

3.3.- There is a considerable missing of data at follow-up (FU). Concretely, 420 PD males were analyzed at baseline and 294 at FU (116 PD males lost to follow-up, 28.3%). 271 PD females at baseline and 202 at FU (69 dropouts, 25.5%). This should be stated in results. Lost to follow-up is a problem in clinical research that it is difficult to handle. However, I think it would be great to know if patients that were lost at FU were different at baseline from those who were not, and write this shortly in results.

AUTHORS – Thank you for your comment. We have conducted the analysis and included this paragraph in Results: “Finally, after comparison of all data of baseline visit (variables included in table 1 and table 2) between patients who were assessed at V24M (N=496) and those patients who lost de follow-up visit at V24M (N=185), no significant differences were detected (only p<0.005 in the PDSS total score, with a worse quality of sleep in patients who lost the follow-up; 108.87 ± 32.28 vs 117.02 ± 24.47; p=0.019). Moreover, in Discussion, this aspect has been updated: “For some variables, the information was not collected in all cases and the data in the follow up was obtained in 496 patients of 681 initially included in the baseline analysis (72.8%), so it could influence the results. However, this percentage is even higher in other longitudinal prospective studies [75] and a bias of withdrawals of more affected patients was excluded after comparing subjects who lost the follow-up vs those who didn´t”.

3.4.- If authors used age at baseline and age at FU in GLM, I would suggest them to reconsider using only age at baseline in the analyses (I am not sure from the text if one or the other was used). If only age at baseline was used, write it accordingly in lines 163, line 265…

AUTHORS – Thank you for your comment. As we commented before, only age at baseline was included. The sentence has been rephrased: “Age at baseline, and levodopa equivalent daily dose (LEDD [32]) at baseline and at V24M were included as covariates”.

Some minor comments:

4.- In Tables, it would be clearer if the first column was center at left and using a different indentation for titles and subtitles.

AUTHORS – Thank you very much for your comment. We don't know if there is a problem with the word processor but it already appears as you say. In any case, we have added (Tables 2, 3 and 4) some differentiating fields to the left, thanks.

5.- As there is a lot of information in Tables, it would help a lot if authors could highlight somehow the significant p-valued (in bold or using asterisks after p-values, for example).

AUTHORS – Thank you very much for your comment. Now, significant p values appears in bold.

6.- Line 131, “except in” seems more appropriate that “apart from”.

AUTHORS – Thank you very much for your comment. It has been corrected.

7.- Standard deviations should be incorporated in Figure 3.

AUTHORS – Thank you very much for your comment. It has been done.

8.- Line 254, “Collectively, the NMS burden (NMSS total score) impaired after the 2-year follow-up in both groups”  I would suggest to change it for “collectively, we observed an impairment of NMS burden (NMSS total score) after…”.

AUTHORS – Thank you very much for your comment. It has been changed.

9.- Discussion. The authors state that LEDD is lower in women, but it has been demonstrated that this is not true after correcting for BMI. If the field is moving towards personalized medicine and sex differences should be considered, the drug posology should also be adapted to BMI.

AUTHORS – Thank you for your comment. Unfortunately, we don´t have this data because weight was collected but not the height. We have added this point in Discussion as a limitation: “The medication dose was not adjusted for the body mass index because this data was not collected”.

10.-Results and discussion. According to Table 4 , there are not differences in the progression of NMS between males and females. This is a very interesting finding that it is not discussed in the text.

AUTHORS – Thank you for your comment. It is commented in Discussion: “Some studies suggest a greater NMS burden in males [62]. In particular, Picillo et al. [63] observed that men complained of a greater number of NMS as compared to women in a 2-year follow-up study as well. Here, we detected only a trend of significance after applying the Bonferroni test (p=0.012) for a greater NMS burden increase in males after a 2-year follow-up, without significant differences by sex in the change of the score of all domains of the NMSS”.

11.- As there is a dropout rate higher than 20%, a sentences stating how missing data could influence the current results is worthy.

AUTHORS – Thank you very much for your comment. An analysis comparing subjects who completed the follow-up vs those who didn´t, showed no differences. It appears in Results: “Finally, after comparison of all data of baseline visit (variables included in table 1 and table 2) between patients who were assessed at V24M (N=496) and those patients who lost de follow-up visit at V24M (N=185), no significant differences were detected (only p<0.005 in the PDSS total score, with a worse quality of sleep in patients who lost the follow-up; 108.87 ± 32.28 vs 117.02 ± 24.47; p=0.019)”. Moreover, it is commented in Discussion: “For some variables, the information was not collected in all cases and the data in the follow up was obtained in 496 patients of 681 initially included in the baseline analysis (72.8%), so it could influence the results. However, this percentage is even higher in other longitudinal prospective studies [75] and a bias of withdrawals of more affected patients was excluded after comparing subjects who lost the follow-up vs those who didn´t”.

Reviewer 2 Report

In the reviewed paper, authors aimed at evaluating the differences between male and female PD patients for a variety of motor and non-motor symptoms as well as several functional outcome measures in a large cohort of Spanish PD patients. Authors found that a significant difference in PD phenotype exists between sex. Although several evidence already exists, the paper is of high interest as gender and precision medicine are a hot topic of research and clinical practice and the study included a large cohort undergoing a very extensive evaluation for a prolonged period of time.

The article is generally well written, clear and concise. Introduction provides a background and justify the aims, methods are clearly described, results are valuable, add information to the field and are reported in a exhaustive and ordinate fashion. Findings are discussed adequately and conclusions are supported by the data.

However a few points for improving methods description and how results are reported and discussed could be addressed:

1)     Please include the reference in the methods (it is already in the reference list, number 24) for NMS score manipulation from simple score to centiles.

2)      Please use the abbreviation MDS-UPDRS if the 2008 scale revision by Goetz and collegues was used (UPDRS is the classical abbreviation for the original 1991 version) and consider adding the reference to the publication (Goetz CG, Tilley BC, Shaftman SR, et al. Movement Disorder Society-sponsored revision of the Unified Parkinson's Disease Rating Scale (MDS-UPDRS): scale presentation and clinimetric testing results. Mov Disord. 2008;23(15):2129-2170. doi:10.1002/mds.22340). If the 1991 version was used, please specify it in the methods.

3)     Please specify in the statistical analysis that the GLM repeated measure design included both a within subjects factor (time/visit) and a between subjects factor (sex), since the description seems to indicate a simple repeated measure design without testing for interaction between these two factors, whereas, in the results, is it clear that a mixed design was applied.

4)     Did you include any means to control family-wise error rate for repeated testing (several GLM analysis were performed, as well as several comparisons at baseline)?

5)     Regarding results, I would recommend against reporting discrete variables (e.g. number of drugs) and semiquantitative ordinal scoring (MDS-UPDRS, etc.) as mean, and use medians instead. Indeed, authors used median to report Hohen and Yahr scale, why did they use a different reporting methods for other semi-quantitative scales?

6)     It could be argued that a statistically significant difference, due to the large sample size, could not be equivalent to a meaningful clinical change. I would discuss this aspect more extensively or include it in the limitations. For instance, the reported difference in motor subpart of MDS-UPDRS between males and females is around 1.1. Minimum clinical difference for MDS-UPDRS part III has been shown to be between 2.5 and 3.25 (see Horváth K, Aschermann Z, Ács P, Deli G, Janszky J, Komoly S, et al. Minimal clinically important difference on the Motor Examination part of MDS-UPDRS. Parkinsonism Relat Disord. (2015) 21:1421–6. doi: 10.1016/j.parkreldis.2015.10.006). Similarly, even if statistically significant, I would consider a difference of 0.92 in rigidity subitems (for instance) not really impactful (bearing in mind, again, that these are semi-quantitative ordinal data and mean values are potentially inappropriate in describing them).

7)      I would include an index of variable dispersion (such as Standard Error bars) in figure 3. This could also help the reader to interpret the significant or non-significant differences between the two groups.

8)      I would avoid using the term “toxic habits” (line 300) as it is misleading and could be misinterpreted. I would opt for something like “harmful” or similar and consider specifying what are the authors including in this definition as it was not used before.

 9)      ICB abbreviation (line 151) is not defined and is not used again the paper

 10)   Nicolletti et al. should be corrected to Nicoletti et al. (line 347)

 11)   I would mention among the limitations that the observational nature of the study does not provide support for cause-effect relationship but can only suggest correlation between variables.

I would like to thank the authors for the valuable, relevant and well conducted study.

Author Response

Reviewer 2

In the reviewed paper, authors aimed at evaluating the differences between male and female PD patients for a variety of motor and non-motor symptoms as well as several functional outcome measures in a large cohort of Spanish PD patients. Authors found that a significant difference in PD phenotype exists between sex. Although several evidence already exists, the paper is of high interest as gender and precision medicine are a hot topic of research and clinical practice and the study included a large cohort undergoing a very extensive evaluation for a prolonged period of time.

The article is generally well written, clear and concise. Introduction provides a background and justify the aims, methods are clearly described, results are valuable, add information to the field and are reported in a exhaustive and ordinate fashion. Findings are discussed adequately and conclusions are supported by the data.

However a few points for improving methods description and how results are reported and discussed could be addressed:

  • Please include the reference in the methods (it is already in the reference list, number 24) for NMS score manipulation from simple score to centiles.

AUTHORS – Thank you very much for your comment. It has been added.

  • Please use the abbreviation MDS-UPDRS if the 2008 scale revision by Goetz and collegues was used (UPDRS is the classical abbreviation for the original 1991 version) and consider adding the reference to the publication (Goetz CG, Tilley BC, Shaftman SR, et al. Movement Disorder Society-sponsored revision of the Unified Parkinson's Disease Rating Scale (MDS-UPDRS): scale presentation and clinimetric testing results. Mov Disord. 2008;23(15):2129-2170. doi:10.1002/mds.22340). If the 1991 version was used, please specify it in the methods.

AUTHORS – Thank you very much for your comment. Unfortunately, in the COPPADIS project the classic scale (UPDRS) was used instead of the new scale (MDS-UPDRS). The methodology can be consult in the link provided about this article: Santos-García D, Mir P, Cubo E, et al.; COPPADIS Study Group. COPPADIS-2015 (COhort of Patients with PArkinson's DIsease in Spain, 2015), a global--clinical evaluations, serum biomarkers, genetic studies and neuroimaging--prospective, multicenter, non-interventional, long-term study on Parkinson's disease progression. BMC Neurol 2016;16:26. In Methods, it is clearly described: “(1) motor assessment (Hoenh & Yahr [H&Y], Unified Parkinson's Disease Rating Scale [UPDRS] part III and part IV…”, and not MDS-UPDRS. In any case and as you suggested, we add a reference about the scale used (reference 19; Fahn S, Elton RL, Members of the UPDRS Development Committee. Unified Parkinson's Disease Rating Scale. In: Fahn S, Marsden CD, Calne DB, Goldstein M, editors. Recent developments in Parkinson's disease, Vol 2. Florham Park, NJ: Macmillan Health Care Information;1987.p-153-64).

  • Please specify in the statistical analysis that the GLM repeated measure design included both a within subjects factor (time/visit) and a between subjects factor (sex), since the description seems to indicate a simple repeated measure design without testing for interaction between these two factors, whereas, in the results, is it clear that a mixed design was applied.

AUTHORS – Thank you very much for your comment. This aspect has been clarified in Methods: “General linear model (GLM) repeated measure were used to test for changes in various scores (motor; NMS; QoL; autonomy for ADL) over time (V24M vs V0 and/or V24M vs V12M and V12M vs V0) separately in each group (men vs women) and test for differences between groups over time. Age at baseline, and levodopa equivalent daily dose (LEDD [32]) at baseline and at V24M were included as covariates. In the latter models, an interaction for visit and group was tested before testing for a group difference over time”.

  • Did you include any means to control family-wise error rate for repeated testing (several GLM analysis were performed, as well as several comparisons at baseline)?

AUTHORS – Thank you for your comment. Family-wise error rate was 1 – (1-0.05)2 = 0.09 (the probability of getting a type I error was 9%). We applied the Bonferroni correction, with α = 0.05 / 25 = 0.002. Family-wise error rate was 0.0039 (the probability of getting a type I error was 0.39%).

  • Regarding results, I would recommend against reporting discrete variables (e.g. number of drugs) and semiquantitative ordinal scoring (MDS-UPDRS, etc.) as mean, and use medians instead. Indeed, authors used median to report Hohen and Yahr scale, why did they use a different reporting methods for other semi-quantitative scales?

AUTHORS – Thank you for your comment. In quantitative scales such as the MDS-UPDRS, the data is usually shown as mean ± standard deviation in the literature. We use the median on the H&Y because this is a scale with only 5 category options (ordinal scale). This point has been commented by other reviewers of articles from the COPPADIS cohort, considering this way to be the most appropriate.

  • It could be argued that a statistically significant difference, due to the large sample size, could not be equivalent to a meaningful clinical change. I would discuss this aspect more extensively or include it in the limitations. For instance, the reported difference in motor subpart of MDS-UPDRS between males and females is around 1.1. Minimum clinical difference for MDS-UPDRS part III has been shown to be between 2.5 and 3.25 (see Horváth K, Aschermann Z, Ács P, Deli G, Janszky J, Komoly S, et al. Minimal clinically important difference on the Motor Examination part of MDS-UPDRS. Parkinsonism Relat Disord. (2015) 21:1421–6. doi: 10.1016/j.parkreldis.2015.10.006). Similarly, even if statistically significant, I would consider a difference of 0.92 in rigidity subitems (for instance) not really impactful (bearing in mind, again, that these are semi-quantitative ordinal data and mean values are potentially inappropriate in describing them).

AUTHORS – Thank you for your comment. As we commented previously, we used the UPDRS instead of the MDS-UPDRS. In the case of the UPDRS-III, we didn´t observed significant differences (p=0.265). This point has been included in Discussion as a limitation: “Another important aspect is that due to the large sample size, some differences observed could be statistically significant but not a minimum clinically important difference (e.g., differences detected in motor aspects such as hypomimia, speech and rigidity) [76], so it is necessary to be cautious about the relevance of some differences detected”.

  • I would include an index of variable dispersion (such as Standard Error bars) in figure 3. This could also help the reader to interpret the significant or non-significant differences between the two groups.

AUTHORS – Thank you for your comment. Standard deviations had been added.

  • I would avoid using the term “toxic habits” (line 300) as it is misleading and could be misinterpreted. I would opt for something like “harmful” or similar and consider specifying what are the authors including in this definition as it was not used before.

AUTHORS – Thank you for your comment. We have changed to harmful habits. Now, the sentence appears as: “… harmful habits (smoking and alcohol consumption), which …”.

  • ICB abbreviation (line 151) is not defined and is not used again the paper.

AUTHORS – Thank you very much for your comment. Regarding your comment, this sentence has been eliminated: “Patients suffering from at least one ICD and/or CB were considered as patients presenting impulse ICB”.

10) Nicolletti et al. should be corrected to Nicoletti et al. (line 347).

AUTHORS – Thank you very much for your comment. It has been corrected.

  • I would mention among the limitations that the observational nature of the study does not provide support for cause-effect relationship but can only suggest correlation between variables.

AUTHORS – Thank you very much for your comment. It has been added in Discussion: “Moreover, the observational nature of the study does not provide support for cause-effect relationship but can only suggest correlation between variables”.

I would like to thank the authors for the valuable, relevant and well conducted study.

AUTHORS – Thank you very much for your comment and time reviewing the manuscript. We are delighted to receive these words from you.
